# A Cross-Stage Partial Network and a Cross-Attention-Based Transformer for an Electrocardiogram-Based Cardiovascular Disease Decision System

**DOI:** 10.3390/bioengineering11060549

**Published:** 2024-05-29

**Authors:** Chien-Ching Lee, Chia-Chun Chuang, Chia-Hong Yeng, Edmund-Cheung So, Yeou-Jiunn Chen

**Affiliations:** 1Department of Anesthesia, An Nan Hospital, China Medical University, Tainan City 709, Taiwan; otzison@gmail.com (C.-C.L.); common0322@gmail.com (C.-C.C.); edmundsotw@gmail.com (E.-C.S.); 2Department of Medical Sciences Industry, Chang Jung Christian University, Tainan City 709, Taiwan; 3Department of Electrical Engineering, Southern Taiwan University of Science and Technology, Tainan City 71005, Taiwan; steve10624@gmail.com

**Keywords:** cardiovascular disease, cross-stage partial network, cross-attention-based transformer, electrocardiograms, cardiovascular disease decision system

## Abstract

Cardiovascular disease (CVD) is one of the leading causes of death globally. Currently, clinical diagnosis of CVD primarily relies on electrocardiograms (ECG), which are relatively easier to identify compared to other diagnostic methods. However, ensuring the accuracy of ECG readings requires specialized training for healthcare professionals. Therefore, developing a CVD diagnostic system based on ECGs can provide preliminary diagnostic results, effectively reducing the workload of healthcare staff and enhancing the accuracy of CVD diagnosis. In this study, a deep neural network with a cross-stage partial network and a cross-attention-based transformer is used to develop an ECG-based CVD decision system. To accurately represent the characteristics of ECG, the cross-stage partial network is employed to extract embedding features. This network can effectively capture and leverage partial information from different stages, enhancing the feature extraction process. To effectively distill the embedding features, a cross-attention-based transformer model, known for its robust scalability that enables it to process data sequences with different lengths and complexities, is employed to extract meaningful embedding features, resulting in more accurate outcomes. The experimental results showed that the challenge scoring metric of the proposed approach is 0.6112, which outperforms others. Therefore, the proposed ECG-based CVD decision system is useful for clinical diagnosis.

## 1. Introduction

Cardiovascular disease (CVD) is a leading cause of death globally, not only impacting patient mortality but also affecting their quality of life and potentially leading to other complications that can jeopardize the health of vital organs. Additionally, CVD patients require significant time and healthcare resources, making early diagnosis essential for reducing harm and medical costs. Using electrocardiography (ECG) to diagnose CVD is the fastest and most convenient method for diagnosing CVD. However, accurate interpretation of the ECG requires extensive professional training and experience [1]. Therefore, developing automatic CVD decision systems for clinical assistance can improve the efficiency of CVD diagnosis and significantly alleviate the burden on healthcare systems.

CVD, such as coronary artery disease, arrhythmia, valvular heart disease, coronary artery heart disease, cerebrovascular disease, rheumatic heart disease, and other related diseases [2], not only affects the cardiovascular system but can also lead to other complications that jeopardize the health of vital organs. Thus, CVD poses a significant threat to human life by not only reducing the lifespan of patients but also impacting their quality of life. According to the World Health Organization, CVD claims the lives of 17.9 million people globally each year. Additionally, according to statistics from the Ministry of Health and Welfare in Taiwan in 2022, heart disease ranks second, causing approximately 23,000 deaths, following malignant tumors, which ranked first [3,4]. These statistics demonstrate the profound impact of CVD on individual health and healthcare resources, highlighting the importance of developing CVD detection systems to reduce harm, improve quality of life, and minimize the demand for healthcare resources.

Recently, CVD can be detected by using cardiac catheterization, echocardiography, and ECG. Cardiac catheterization is the most invasive and high-risk approach [5], usually reserved for cases where other methods cannot detect the disease or for cardiovascular disease treatment purposes. Echocardiography can be invasive or non-invasive [6,7], with a measurement time of at least 15 min. On the other hand, ECG is a non-invasive measurement method [8], requiring only the attachment of electrodes to the subject, and can be completed within five minutes. Therefore, using ECG is the most convenient method for detecting CVD. However, accurate interpretation of CVD using ECG requires extensive professional training and experience. Hence, developing intelligent CVD decision systems to assist professionals in accurately interpreting CVD through ECG can enhance their training and reduce their workload.

In recent years, deep learning techniques have been widely and successfully applied in various fields, demonstrating the high practicality of deep learning. Therefore, using deep learning techniques in CVD decision systems can effectively improve the recognition rate. In the context of CVD applications, PhysioNet and Cardiovascular Disease Computing jointly organized the PhysioNet Challenge to promote the technological development of CVD decision systems [9,10]. Zhu et al. used the SE-ResNet residual neural network architecture to detect CVD [11]. SE-ResNet can effectively enhance the feature extraction capability by adapting the importance of different feature channels based on learning the correlations between features. Zhao et al. proposed a deep neural network architecture that combined an improved ResNet with an SE layer [12]. The SE layer can model the spatial relationship between channels, and the improved ResNet can effectively learn features from time series data. Natarajan et al. developed a broad and deep transformer neural network for CVD classification [13]. This approach adopts a transformer neural network to learn discriminative feature representations from each 12-lead ECG sequence. Racha et al. used ResNet-type architectures, and the proposed architecture can effectively learn from shorter ECG segments [14]. Therefore, using novel neural network architectures can help develop CVD detection systems.

Neural networks have been applied to various fields [15,16,17,18]. The researchers showed that neural networks with deeper and broader architectures can significantly improve system performance. However, this would also require more parameters and computational resources. To address this issue, Wang et al. proposed a cross-stage partial network (CSPNet) to reduce the impact of parameters and computational resources [15]. Therefore, Ali et al. proposed GPA-Net, a neural network architecture that includes CSPNet, CTA, and SPA, and successfully developed a pest detection system based on the Internet of Things technology [16]. Hao et al. used CSPNet to design a lightweight convolutional neural network [17] for a synthetic aperture radar image ship target detection system. In this study, the developed neural network architecture has feature fusion capabilities to reduce model parameters and improve system accuracy. Ju et al. proposed the Graph-CSPNet architecture for developing a brain–computer interface based on motor imagery [18]. This neural network architecture utilizes graph convolution techniques to capture electroencephalography features in the time–frequency domain, enhancing the signal segmentation ability of local fluctuations and improving the recognition rate of the system. Therefore, integrating CSPNet into the CVD decision system can effectively reduce the complexity of the network architecture and then improve system performance.

The transformer networks have efficiency advantages, capturing long-range dependencies, extracting global information, flexibility in adjustment, and good generalization ability. Therefore, transformer networks have been widely used in various tasks, such as natural language processing, machine translation, sentiment analysis, and image processing [11,19,20,21,22]. Natarajan et al. [11], Li et al. [19], and Qiu et al. [20] have designed transformer network architectures for ECG-based applications, and the experimental results showed that the transformer network has high performance and is valid for practical applications. Thus, using the transformer network can improve the performance of the CVD decision system.

Cross-attention neural networks are developed to effectively capture the relationships between different input data [23,24,25,26,27,28], and then an enhanced feature representation can be extracted. Huang et al. proposed a novel cross-attention module that collects contextual information from all pixels along its cross-attention paths. Then the GPU memory usage is reduced by 11 times [23]. Chen et al. presented a dual-branch vision transformer for learning multi-scale features [24]. The proposed neural network is based on a cross-attention network, and a fusion method based on cross-attention is developed to efficiently exchange information between two branches in linear time. Recently, self-attention and cross-attention have been applied to the proposed intelligent systems [25,26], and the results showed that their method achieved better precision than other approaches. In addition, Lin et al. [27] and Huo et al. [28] use cross-attention mechanisms to enhance multi-scale feature maps in transformer-based neural networks, and then the computational overhead can be greatly reduced. Thus, for a CVD decision system, the cross-attention mechanism is very suitable for clinical application because it reduces computational resources.

In this study, an ECG-based CVD decision system with CSPNet and a cross-attention mechanism is proposed for clinical assistance and used to alleviate the burden on healthcare systems. To effectively reduce the complexity of the neural network architecture, CSPNet is adopted for feature extraction. To find a precise embedding feature representation, a transformer neural network is used to capture long-range dependencies and extract global information. To reduce computational resources, the cross-attention mechanism is integrated with the transformer neural network. Finally, a decision result is precisely identified using a multilayer perceptron network.

The rest of this paper is organized as follows: The proposed ECG-based CVD decision system with CSPNet and a cross-attention-based transformer is described in Section 2. Section 3 then conducts a series of experiments to evaluate the performance of our approach. Conclusions and recommendations for future research are finally drawn in Section 4.

## 2. The ECG-Based Cardiovascular Disease Decision System

The proposed neural network architecture for the CVD decision system with CSPNet and a cross-attention-based transformer is presented in Figure 1. First, a CSPNet is designed for feature extraction, and an embedding feature can be automatically extracted to represent an input ECG signal. Second, a transformer with a cross-attention mechanism is designed to effectively distill the embedding features to find a meaningful embedding feature and to reduce computational resources. Finally, a multilayer perceptron network is adopted to find the final decision. Each neuron in the multilayer perceptron network (FC) is connected to all neurons in the previous layer, and it follows the feedforward artificial neural network method. This process is described in detail in the following.

### 2.1. CBM Neural Network

A CBM neural network contains a convolutional layer, a batch normalization, and a mish activation function, and the neural network architecture is shown in Figure 2. The convolutional layer convolves the input, which is the output of previous layers, and passes its results as the output. A batch normalization process is used to normalize the layers’ inputs by re-centering and re-scaling to make the training of artificial neural networks faster and more stable. For an input of batch normalization, *x*, the operation of batch normalization, *BN*(∙), is defined as
(1)$$\mathrm{BN}\left(x\right)=\gamma \frac{x-\mu }{\sqrt{\sigma^{2}+\varepsilon }}+\beta ,$$
where *μ* and *σ* are the per-dimension mean and standard deviation, respectively. *ε* is added in the denominator for numerical stability and is an arbitrarily small constant. *γ* and *β* are the transformation parameters subsequently learned in the optimization process.

The mish function is used as a smooth approximation of the rectifier, and it is defined as
(2)$$\mathrm{Mish}\left(x\right)=x\tanh\left(\log\left(1+e^{x}\right)\right),$$
where *tanh*(∙) is the hyperbolic tangent.

### 2.2. Cross-Stage Partial Network

The CSPNet is a variant of the ResNet architecture, and it can achieve a richer gradient combination while reducing the amount of computation. The neural network architecture of the designed CSPNet is shown in Figure 3. *N* is the number of the used ResUnit, which is a residual network. The ResUnit, in which the weight layers learn residual functions concerning the layer inputs, and the architecture are shown in Figure 4. Moreover, the residual network can be easily trained to obtain better accuracy. Thus, for the proposed CSPNet, the residual network is selected as the ResUnit, composed of two CBM units. Moreover, the output of the ResUnit is added to the input by using element-wise addition.

In the CSPNet, dropouts are adopted to reduce overfitting in neural networks. At each training stage, individual nodes are removed with a predefined probability, and the reduced network is trained on the data during that stage.

### 2.3. Cross-Attention-Based Transformer

The proposed cross-attention-based transformer includes a transformer unit and a cross-attention unit, and the neural network architecture is shown in Figure 5. The cross-attention-based transformer contains no recurrence and no convolution. To use the order of the sequence, some information about the relative or absolute position of the tokens in the sequence is injected into the proposed architecture. Moreover, the positional encodings have the same dimension, *d_m_*, as the input embeddings, so the input embeddings and the positional encodings can be summed. In this study, the sine and cosine functions of different frequencies are selected as the positional encodings, *PE*, and defined as
(3)$$PE_{\left(pos,2i\right)}=\sin\left(\frac{pos}{1000^{\frac{2i}{d_{m}}}}\right),$$
and
(4)$$PE_{\left(pos,2i+1\right)}=\cos\left(\frac{pos}{1000^{\frac{2i}{d_{m}}}}\right),$$
where *pos* and *i* are the position and the dimension, respectively. Therefore, each dimension of the positional encoding corresponds to a sinusoid, and the wavelengths form a geometric progression from 2π to 10,000 · 2π.

The transformer unit has two sub-layers. The first is a multi-head self-attention mechanism, and the second is a fully connected feed-forward network. The residual connection is around the two sub-layers, followed by layer normalization.

The cross-attention unit is a kind of multi-head attention, and the cross-attention combines asymmetrically two separate embedding sequences of the same dimension. In this study, the inputs of multi-head attention are the outputs of the previous layer, *a_p_*, and the far layer, *a_f_*. The *a_p_* is multiplied by the weight matrix *w_q_*, and the *a_f_* is multiplied by the weight matrix *w_k_* and *w_v_*. The weight matrixes are trained in the training process, and then the query vector *Q_i_*, the key vector *K_i_*, and the value vector *V_i_* can be obtained as
(5)$$Q_{i}=w_{q}a_{p},$$
(6)$$K_{i}=w_{k}a_{f},$$
and
(7)$$V_{i}=w_{v}a_{f}.$$

When the *Q_i_*, *K_i_*, and *V_i_* are obtained, the attention operation *Attention*(∙), which is modeled as dot-production attention, is used to find the weighted attention outputs *SA_i_*. *Attention*(∙) is defined as
(8)$$\begin{array}{c} SA_{i}=Attention\left(Q_{i},K_{i},V_{i}\right)\\ \;\;=\mathrm{softmax}\left(\frac{Q_{i}K_{i}^{T}}{\sqrt{d}}\right)•V_{i} \end{array}$$
where softmax(∙), *T*, and *d* are the softmax function, the transpose operation, and the scaling factor, respectively.

## 3. The Experimental Results and Discussions

To evaluate the proposed approaches, the dataset in PhysioNet/Computing in Cardiology Challenge 2020 [10] is used, and the detailed results are detailed in the following subsections.

### 3.1. Dataset and Evaluation Metric

The datasets for this Cardiology Challenge are from the CPSC database and CPSC-Extra database [29], the INCART database [10], the PTB and PTB-XL databases [30], the Georgia 12-lead ECG Challenge (G12EC) database [9], and an undisclosed database. Therefore, the numbers of ECG signals for CPSC, CPSC-Extra, INCART, PTB, PTB-XL, and Georgia are 6877, 3453, 74, 516, 21,837, and 10,344, respectively. The sampling rate is normalized at 500 Hz, and the number of classes for cardiology diseases is 27. Moreover, the K-fold cross-validation technique is adopted to evaluate the proposed approaches. In this study, the dataset is randomly divided into ten folds, and then the number of folds for training and testing is nine and one, respectively. In the database, each subject almost has only one record, so in the case of the k-fold validation methodology, the subjects for training and testing are entirely different. In terms of experimental design, on one hand, to objectively compare with existing systems, we followed the experimental design in PhysioNet/Computing in Cardiology Challenge 2020 [10] for k-fold validation and did not discuss using different datasets for training and testing.

In this study, we selected a challenge scoring metric (CM) developed for the PhysioNet/Computing in Cardiology Challenge 2020 to evaluate our proposed approach [10]. CM was chosen because it reflects the clinical reality that some misdiagnoses can be more harmful than others and should be scored accordingly. Additionally, it considers the fact that confusing certain classes is less detrimental than confusing others.

Moreover, CM assigns partial credit to misdiagnoses that result in similar treatments or outcomes as the true diagnosis, as judged by cardiologists. This means that CM gives full credit to correct diagnoses while providing partial credit to misdiagnoses that have similar risks or outcomes as the true diagnosis.

Consequently, CM rewards true positives, partially rewards false negatives, and penalizes false positives by not giving any credit or reducing the credit for true positives and false negatives. It is important to note that a classifier that only returns positive outputs will receive a negative score, which means a lower score compared to a classifier that only returns negative outputs.

Moreover, the hyperparameters of the proposed approaches are shown in Table 1. The Adam algorithm with *β* = 0.9, *β*_2_ = 0.98, and *ε* = 10^−9^ was selected as the optimizer for the training neural networks. The number of iterations and batch size were 30 and 128, respectively.

### 3.2. The Results of the Transformer with Cross-Attention

In this subsection, the effect of the cross-attention model with different architectures is examined. The transformer without the cross-attention model is selected as the baseline and denoted as Baseline. Two transformer models with two different structures of cross-attention models (denoted as CAT1 and CAT2) shown in Figure 6 were compared with the proposed approach. The proposed cross-attention model used in the transformer model is to fuse the variant information of different neural layers. Compared with the proposed approach, CAT1 is designed to fuse the embedding features obtained from the third layer, which is deeper than the proposed approach. Additionally, CAT2 is designed to fuse variant embedding features obtained from multiple neural layers.

The experimental results are shown in Table 2. In Table 2, it is clear that the transformer with the cross-attention model outperforms the baseline system. Therefore, the cross-attention model can effectively integrate different information between different layers, and then the performance of the CVD system can be improved. Moreover, the proposed architecture with the cross-attention model can achieve the highest score. Compared with the proposed approach and CAT1, the embedding features in the second layer are more useful than those of the third layer. In CAT2, more embedding features of different layers are fused, but the performance is not improved and is even lower than the proposed approach. Therefore, selecting a suitable architecture for fusing different embedding features can improve the accuracy of the CVD system. In this study, only trying different combinations to find the optimal architecture is time-consuming. Therefore, developing an optimization method to design neural network architectures can effectively reduce the system development cycle.

### 3.3. The Results of CSPNet

In this subsection, the different architectures for using CSPNet are examined and shown in Figure 7. The architectures using sequence and parallel structures are denoted as CSP1 and CSP2, and the experimental results are shown in Table 3. The mean and standard variance of the proposed approaches, CSP_S, CSP_P1, and CSP_P2, are 0.6112 ± 0.0201, 0.5893 ± 0.0225, 0.6000 ± 0.0258, and 0.6043 ± 0.0208, respectively. The results showed that the parallel structure can extract more useful information, and the performance can be improved. Comparing CSP_P1 and CSP_P2, the number of branches for a parallel structure is the same, but the level of CSP_P2 is deeper than that of CSP_P1. The performance can, therefore, be slightly improved. Thus, having a deep layer for neural networks is very important for performance improvement. Balancing performance and computational complexity is an important issue when designing the structure of neural networks. Therefore, the proposed approach reduces CSPNet by replacing a parallel structure with a sequential structure. By combining sequential and parallel structures, the proposed approach can achieve the highest score. Thus, selecting a suitable structure for using CSPNet can effectively improve the performance.

### 3.4. The Results Compared with Other Approaches

In this subsection, we will select the most recent system that uses the same database as the baseline system for comparison. The top 10 systems from the PhysioNet/Computing in Cardiology Challenge 2020 have been listed in this study [10]. From these, the PRNA system, which is ranked first, was selected as the baseline system for comparison. Furthermore, in recent years, the database used in this study has also been used by other systems based on transformer neural networks and ResNet neural networks. Therefore, we also compared the results with the study that outperformed the PRNA system by using these transformer and ResNet-based systems as baselines.

In this study, ResNet transformer [14], PRNA [8], Weighted ResNet [12], and SE-ResNet [11] were selected as the baseline systems and compared with proposed approaches. The experimental results for the ResNet transformer, PRNA, Weighted ResNet, and SE-ResNet are 0.6080 ± 0.0108, 0.5331 ± 0.0464, 0.520, and 0.514, respectively. The results of the proposed approach and ResNet transformer, based on transformer-based neural networks, outperform PRNA, ResNet, and SE-ResNet. The transformer-based neural network can effectively distill the embedding features, compared with only using ResNet. Moreover, CSPNet can precisely extract embedding features from ECG signals, compared with deep convolutional neural networks.

## 4. Conclusions

In this study, an ECG-based CVD decision system with CSPNet and a cross-attention-based transformer is proposed to alleviate the burden on healthcare systems. The CSPNet is adopted as the feature extraction, and the extracted embedding can precisely represent the input ECG signals. The transformer with cross-attention can effectively distill the embedding features and reduce computational resources. The experimental results showed that the proposed approach outperforms other approaches. Therefore, the proposed approach can improve the efficiency of CVD diagnosis and alleviate the burden on healthcare systems. In the future, the number of parameters can be greatly reduced by using the teacher–student model, and then it would be helpful for practical applications.

## Figures and Tables

**Figure 1 bioengineering-11-00549-f001:**
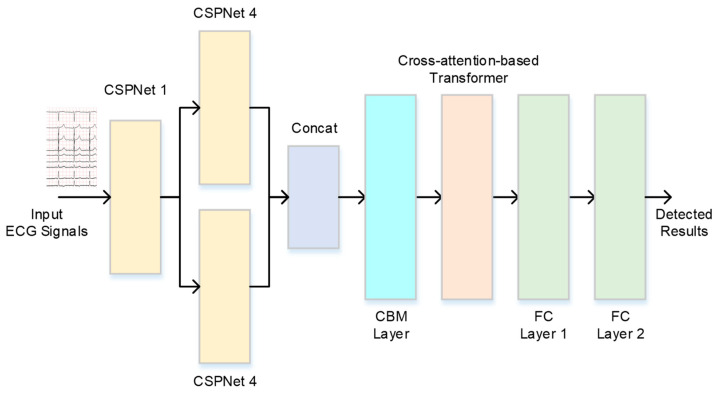
The architecture of the proposed CVD decision system.

**Figure 2 bioengineering-11-00549-f002:**
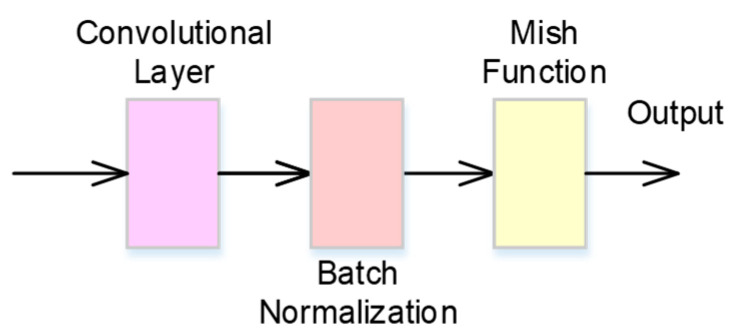
The architecture of the CBM neural network.

**Figure 3 bioengineering-11-00549-f003:**
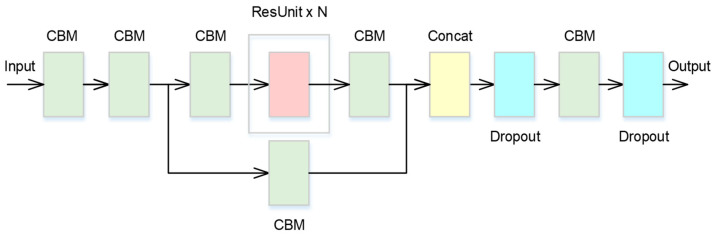
The architecture of the CSPNet N.

**Figure 4 bioengineering-11-00549-f004:**
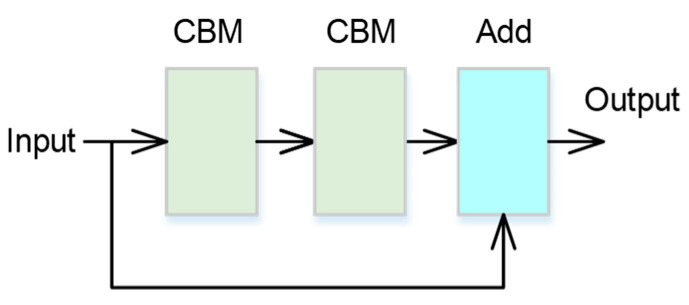
The architecture of the ResUnit.

**Figure 5 bioengineering-11-00549-f005:**
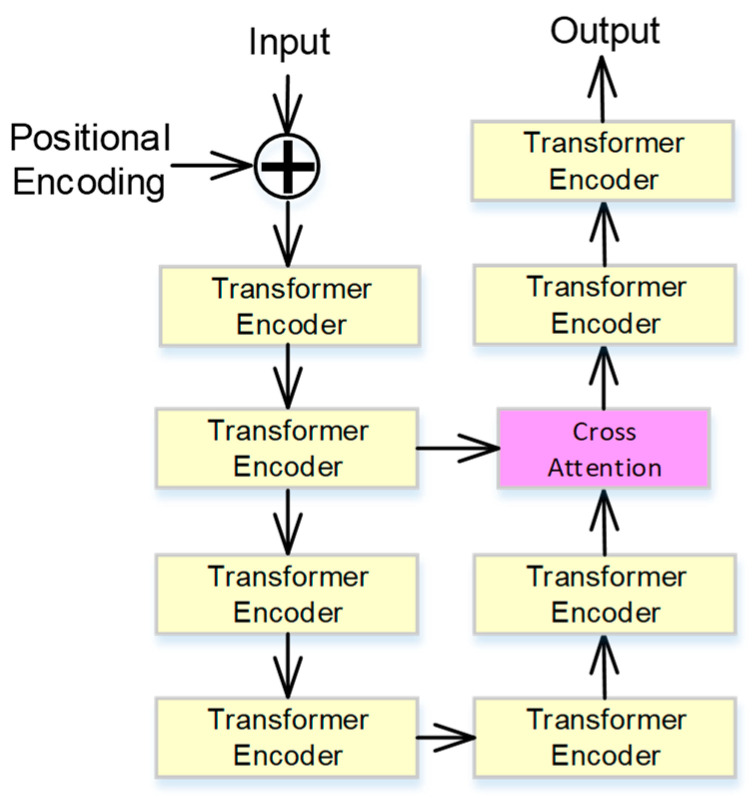
The architecture of the designed cross-attention-based transformer.

**Figure 6 bioengineering-11-00549-f006:**
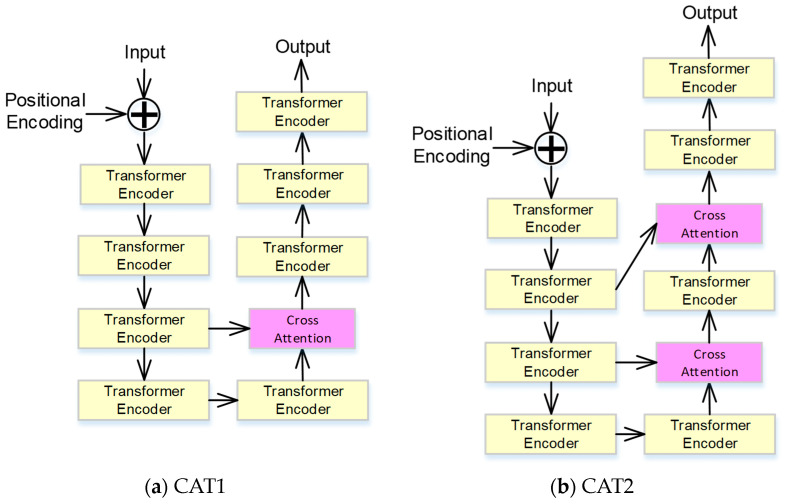
The transformer models with different structures of cross-attention mode.

**Figure 7 bioengineering-11-00549-f007:**
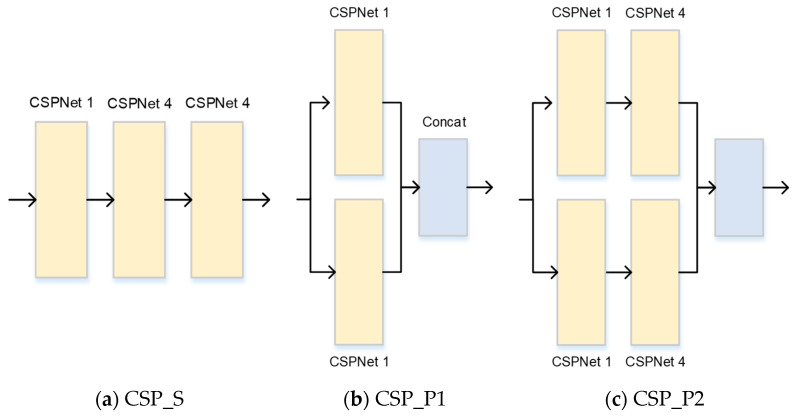
The different architectures of using CSPNet with sequence and parallel structures.

**Table 1 bioengineering-11-00549-t001:** The hyperparameters of the proposed CVD decision system. CS, KS, S, P, and R are the channel number, kernel size, step of stride, padding number, and dropout rate.

Model	Sub-Model	Hyperparameters	Value
CSPNet 1	CBM 1 CBM 2 CBM 3 CBM 4 ResUnit x 1 CBM 5, 6 Dropout	CS, KS, S, P CS, KS, S, P CS, KS, S, P CS, KS, S, P CS, KS, S, P CS, KS, S, P R	32,15,1,7 32,14,2,6 64,10,2,4 32,10,2,4 32,11,1,5 64,11,1,5 0.2
CSPNet 4	CBM 1 CBM 2 CBM 3 CBM 4 ResUnit x 1 CBM 5 CBM 6 Dropout	CS, KS, S, P CS, KS, S, P CS, KS, S, P CS, KS, S, P CS, KS, S, P CS, KS, S, P CS, KS, S, P R	64,15,3,6 32,10,2,4 32,11,1,5 64,11,1,5 64,11,1,5 32,11,1,5 32,14,3,0 0.2
CBM	CBM	CS, KS, S, P	64,15,1,7
Cross- attention- based transformer	Embedding size Head Layers Feedforward Dropout	*d_m_* *N_h_* *N_L_* KS R	64 4 8 2048 0.1
Fully connected layer	FC1 Dropout FC2	KS R KS	64 0.2 27

**Table 2 bioengineering-11-00549-t002:** The experimental results for different architectures with the cross-attention models.

Fold	Proposed Approach	Baseline	CAT1	CAT2
0	0.6272	0.6190	0.5896	0.5959
1	0.6322	0.6175	0.6271	0.6223
2	0.6199	0.6076	0.6121	0.6280
3	0.6324	0.6101	0.6333	0.6324
4	0.6304	0.6194	0.6217	0.6242
5	0.6065	0.6103	0.6207	0.6064
6	0.5919	0.6103	0.5886	0.5935
7	0.6024	0.5925	0.6114	0.6028
8	0.5922	0.5742	0.5745	0.5879
9	0.5765	0.5740	0.5737	0.5805
Avg C.M	0.6112 ± 0.0201	0.6035 ± 0.0173	0.6053 ± 0.0219	0.6074 ± 0.0183

**Table 3 bioengineering-11-00549-t003:** The experimental results for different architectures with the cross-attention models.

Fold	CSP_S	CSP_P1	CSP_P2
0	0.6052	0.6064	0.6379
1	0.6102	0.6283	0.5995
2	0.6126	0.6078	0.6086
3	0.6129	0.6195	0.6213
4	0.5743	0.6196	0.6216
5	0.5835	0.6209	0.6202
6	0.6000	0.5869	0.5827
7	0.5796	0.5881	0.5920
8	0.5463	0.5838	0.5859
9	0.5683	0.5416	0.5737
Avg C.M	0.5893 ± 0.0225	0.6000 ± 0.0258	0.6043 ± 0.0208

## Data Availability

The data presented in this study are available at https://doi.org/10.13026/dvyd-kd57, reference number [10].

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
