# Peer review of "A Cross-Stage Partial Network and a Cross-Attention-Based Transformer for an Electrocardiogram-Based Cardiovascular Disease Decision System"

_bioengineering, 2024, doi:10.3390/bioengineering11060549_

Round 1
Reviewer 1 Report
Comments and Suggestions for Authors
The paper deals with an ECG-based CVD decision system. The results are quite interesting and I think that the paper deserves publication.
I have only some minor remarks, that are listed as follows.
· Abstract: please, revise. Too much acronyms make this summary a bit difficult to read. I suggest to reduce the number of acronyms and add just another sentence about the obtained results.
· Formula (2) should be revised. The multiplication by the symbol “x” can be omitted in this case.
· Is it possible to provide some further details about results of subsections 3.1, 3.2 and 3.3?
My final decision is “minor revisions”.
Reviewer 2 Report
Comments and Suggestions for Authors
This paper proposes and evaluates a deep learning architecture for developing a cardiovascular disease decision system. The paper is interesting including several datasets. My main concerns is about the comparison to previous works. I miss a stronger comparison.
Comments to improve the paper:
· At the end of the introduction, I’d suggest including a list of contributions.
· In the experiments, you used a kfold cross validation methodology. I have two questions: did you use a validation subset for finetuning the hyperparameters? Did you consider a subject-wise strategy? ( I mean recording from different subjects were considered for training and testing?
· Have you tried training with one dataset and testing with another?
· Can you extend the description of the scoring metric?
· As the datasets are public, how does you best system compare to sota results on these datasets? I think, it would be interesting presenting a comparison table including number from previous papers.
Reviewer 3 Report
Comments and Suggestions for Authors
Interesting paper using among others transformers and cross-attention. The authors however need to sddress the following issues:
1) The authors state that they use two different cross attention architectures. The authors should clarify the two different architectures that are stated and further explain the different measures as it concerns the evaluation of the two different architectures.
2) The paper lacks a discussion section where limitations of the proposed methodology should be laid out, and a solid interpretation of the changes in the evaluation / classification metrics.
Comments on the Quality of English Language
English is OK.
Round 2
Reviewer 2 Report
Comments and Suggestions for Authors
The authors have addressed my comments. Just one minor aspect, the authors explain that they followed the experimental design in PhysioNet/Computing in Cardiology Challenge, I'd suggest including a reference to see this experimental design.
Author Response
Thank you for your valuable feedback to our manuscript. We have revised the manuscript to address your concerns. We hope that the revised version of the manuscript is in much better shape now.
The authors have addressed my comments. Just one minor aspect, the authors explain that they followed the experimental design in PhysioNet/Computing in Cardiology Challenge, I'd suggest including a reference to see this experimental design.
Responses: Thank you very much for your valuable suggestion. A reference is cited to view the experimental design.
Please refer page 7, line 222:
“In terms of experimental design, on one hand, to objectively compare with existing systems, we followed the experimental design in PhysioNet/Computing in Cardiology Challenge 2020 [10] for k-fold validation and did not discuss using different datasets for training and testing.”
Reviewer 3 Report
Comments and Suggestions for Authors
Authors addressed the revisions.
Comments on the Quality of English LanguageEnglish is ok
Author Response
Thank you for your valuable feedback to our manuscript. We hope that the revised version of the manuscript is in much better shape now.
Authors addressed the revisions.
Response: Thank you very much.